# Fungal Endophytes as Mitigators against Biotic and Abiotic Stresses in Crop Plants

**DOI:** 10.3390/jof10020116

**Published:** 2024-01-30

**Authors:** H. G. Gowtham, P. Hema, Mahadevamurthy Murali, N. Shilpa, K. Nataraj, G. L. Basavaraj, Sudarshana Brijesh Singh, Mohammed Aiyaz, A. C. Udayashankar, Kestur Nagaraj Amruthesh

**Affiliations:** 1Department of Studies and Research in Food Science and Nutrition, KSOU, Mysuru 570006, Karnataka, India; gajendramurthygowtham@gmail.com; 2Department of Studies in Botany, University of Mysore, Manasagangotri, Mysuru 570006, Karnataka, India; mdphema@gmail.com (P.H.); botany.murali@gmail.com (M.M.); knataraj922@gmail.com (K.N.); brijeshrajput.bt@gmail.com (S.B.S.); 3Department of Studies in Microbiology, University of Mysore, Manasagangotri, Mysuru 570006, Karnataka, India; shilpanatraj@gmail.com; 4PG Department of Botany, Maharani’s Science College for Women, JLB Road, Mysuru 570005, Karnataka, India; glbasavarj76@gmail.com; 5Department of Studies in Biotechnology, University of Mysore, Manasagangotri, Mysuru 570006, Karnataka, India; reachaiyaz@gmail.com

**Keywords:** biotic stress, abiotic stress, drought, fungal endophytes, pests, phytopathogens

## Abstract

The escalating global food demand driven by a gradually expanding human population necessitates strategies to improve agricultural productivity favorably and mitigate crop yield loss caused by various stressors (biotic and abiotic). Biotic stresses are caused by phytopathogens, pests, and nematodes, along with abiotic stresses like salt, heat, drought, and heavy metals, which pose serious risks to food security and agricultural productivity. Presently, the traditional methods relying on synthetic chemicals have led to ecological damage through unintended impacts on non-target organisms and the emergence of microbes that are resistant to them. Therefore, addressing these challenges is essential for economic, environmental, and public health concerns. The present review supports sustainable alternatives, emphasizing the possible application of fungal endophytes as innovative and eco-friendly tools in plant stress management. Fungal endophytes demonstrate capabilities for managing plants against biotic and abiotic stresses via the direct or indirect enhancement of plants’ innate immunity. Moreover, they contribute to elevated photosynthesis rates, stimulate plant growth, facilitate nutrient mineralization, and produce bioactive compounds, hormones, and enzymes, ultimately improving overall productivity and plant stress resistance. In conclusion, harnessing the potentiality of fungal endophytes represents a promising approach toward the sustainability of agricultural practices, offering effective alternative solutions to reduce reliance on chemical treatments and address the challenges posed by biotic and abiotic stresses. This approach ensures long-term food security and promotes environmental health and economic viability in agriculture.

## 1. Introduction

The ever-expanding population has escalated the demand for food availability and necessitated the development of newer strategies to feed different life forms. In order to achieve the increasing food demand, it is essential to achieve maximum crop yield and reduce the losses inflicted by biotic (viz., phytopathogens, pests, and nematodes) and abiotic stresses (viz., drought, salinity, heat, and heavy metals) to ensure global food security [1,2,3]. Overusing synthetic chemicals to protect plants against various biotic and abiotic stresses has contributed to deleterious effects on ecosystems (including soil and water pollution) and harmful impacts on non-target organisms, leading to the emergence of chemical resistance [4]. Addressing the challenges posed by these stressors in plants is vital to ensure food security and address economic, environmental, and public health concerns. Sustainable alternative approaches and practices are thus required to minimize the agricultural use of chemicals, diminish their negative impacts, and address the challenge of ensuring long-term food security [5]. Endophytes offer innovative and sustainable methods that could reduce crop losses while minimizing the negative impact of conventional chemical treatments in managing biotic and abiotic stresses [6,7].

Endophytes are fascinating microbes that inhabit plant tissues without harming or causing disease symptoms to their host plants [8]. These organisms are crucial for maintaining plant health, and they contribute to ecosystem functioning by establishing a beneficial/symbiotic association with their host during the partial or entire life cycle [9,10]. Endophytes are variably distributed based on the species of host plants and the environments in which the plants thrive. Most vascular plants host endophytes that are thought to evolve in the phyllosphere and rhizosphere before entering via wounds or natural openings. They are present in a wide variety of plant species (including grasses and trees) as well as in different parts of plants (roots, stems, and leaves). Endophytes can transmit either vertically (systemic) from the parent to offspring or horizontally (non-systemic) between unrelated plants [11]. Endophytes have attracted interest in agricultural, industrial, and biotechnological applications in medicine due to their possible usage in crop improvement, biocontrol activities, enzyme production, bioremediation, nutrient cycling, biodegradation, and biotransformation [12,13,14,15].

Endophytes are a rich source of diverse bioactive metabolites, such as flavonoids, quinones, alkaloids, phenolic acids, saponins, tannins, steroids, and terpenoids, which have shown potential benefits, including antimicrobial, anticancer, anti-inflammatory, antioxidant, anti-malarial, anti-diabetic, anti-tuberculosis, anti-arthritis, and immunosuppressive properties [15,16,17,18,19]. It is important to note that although these bioactive compounds play a crucial role in enhancing the host plant’s stress resistance, they also have the potential for various therapeutic applications [11]. It is well established that the use of fungal endophytes offers a more sustainable and environmentally friendly approach to the management of biotic and abiotic stresses in plants by enabling the host plant to adapt to a particular stress, lowering the possible risks to unintended organisms and reducing the amount of chemical residues in food [20]. The present review focuses on fungal endophytes and their mechanisms in the management of biotic and abiotic stresses in plants. It also discusses recent advances and future directions in studying fungal endophytes and their potential agricultural applications.

## 2. Fungal Endophytes

Endophytes, which include both fungus and bacteria, are microorganisms that live inside the host’s cells and create local or systemic connections that exhibit mutuality, similarly to a symbiotic relationship, without clearly posing a sign of illness. It is commonly recognized that these endophytes increase plants’ capacity to withstand stress and recycle nutrients, increase their fitness, and encourage development. Compared to other endophytic microbes, fungal endophytes have attracted the greatest interest since they are a polyphyletic, ecologically relevant group that includes ascomycetes, basidiomycetes, and anamorphic fungi [21,22]. The interaction between fungal endophytes and their host plants is significantly influenced by the two main transmission patterns (vertical and horizontal) [11]. Vertical transmission describes the fungal endophyte transfer from the maternal plant to its progeny through the penetration of the fungal hyphae into the host seed embryo. These vertically transmitted endophytes frequently contribute to a mutualistic relationship with their host plants and may benefit them by increasing their disease resistance, improving their nutrient uptake, and tolerating environmental stresses [9,10]. During horizontal transmission, fungal endophytes can spread from one plant to another, usually through the sexual spores produced on the fungal stroma [23]. In contrast to vertically transmitted endophytes, these can establish an antagonistic or competitive relationship with their host plant [24].

Fungal endophytes have been established to confer robust biocontrol towards biotic and abiotic stresses in host plants through the direct inhibition of pathogen activity through competition for niches and nutrients, antibiosis, and mycoparasitism, in addition to indirect inhibition via an induced plant resistance mechanism by activating a plant’s defense system [25]. Using fungal endophytes, as bioinoculants, in plant colonization elevates plant productivity through increased photosynthesis rates; the stimulation of plant growth; the mineralization of essential nutrients; enzymatic activities; the production of phytohormones, volatile organic compounds, and bioactive metabolites; and the enhancement of tolerance to both biotic and abiotic stresses [7,19,26,27]. Khalil et al. [28] have revealed the potentiality of fungal endophytes (such as *Alternaria tenuissima*, *Aspergillus flavus*, *Penicillium caseifulvum*, *P. commune*, and *P. crustosum*) isolated from the *Ephedra pachyclada* plant to improve *Zea mays*’ plant defense and growth efficiency and soil quality by secreting bioactive compounds and extracellular lytic enzymes (viz., cellulase, amylase, catalase (CAT), and pectinase), phosphate solubilization, and the production of hormones (viz., indole acetic acid) and ammonia. Fungal endophytes are mostly recognized for their capacity to synthesize a wide range of bioactive metabolites, and therefore, they are also usefully intended to play the dual role of improving plant protection and health by enhancing soil quality and assisting host plants in overcoming biotic and abiotic stresses, thus reducing the usage of chemical fertilizers and pesticides in the agricultural sector [7,29,30,31].

## 3. Role of Fungal Endophytes in the Mitigation of Biotic Stress

Biotic stress, induced by harmful microbes, causes a significant challenge to normal plant growth and has widespread negative effects globally [6]. Various biotic factors, namely microbes, pests, and nematodes, which act as major stressors, lead to the increased production of reactive oxygen species (ROS) that impact both the molecular and physiological processes in plants, ultimately reducing crop productivity and even causing plant death [3]. Fungal endophytes offer a solution to address the challenges associated with conventional farming by colonizing plant tissues without damaging them. Fungal endophytes participate in enhancing the fitness of hosts in response to biotic stress by facilitating nutrient uptake, synthesizing phytohormones, and mitigating injury caused by pathogens through mechanisms such as antibiosis, lytic enzyme production, the biosynthesis of secondary metabolites, mycoparasitism, and the activation of induced systemic resistance (Figure 1).

### 3.1. Role of Fungal Endophytes against Fungal Pathogens

Fungal pathogens significantly impact agriculture because they cause plant diseases that lower crop production and quality [32,33]. In addition, the toxins produced by phytopathogenic fungi play a pivotal role in developing plant diseases, consequently harming host plants [34]. Most phytopathogenic toxins are secondary metabolites with low molecular weights that are capable of inducing specific disease symptoms, including stunted growth, chlorosis, wilting, necrotic spotting, and plant death. These toxins can disrupt normal plant physiological processes, even at very low concentrations [2]. Despite being frequently used to control pathogenic fungi, chemical fungicides have negative environmental effects. Fungal endophytes can significantly modulate the impacts of fungal diseases, reducing the adverse environmental effects of chemical fungicides [35,36,37]. Fungal endophytes use diverse mechanisms to control fungal pathogens and induce plant resistance, including activating defense-related gene expression and synthesizing bioactive metabolites and hormonal signaling molecules [38]. It has been well observed that the growth and development of pathogenic fungi can be hampered by the antimicrobial substances that fungal endophytes might produce as secondary metabolites, thereby directly lowering their prevalence and virulence within the host plant [39] (Table 1).

Certainly, some fungal endophytes may possess biocontrol potential through their ability to produce antimicrobial compounds, thereby inhibiting the competition of fungal pathogens for nutrients and space within the host plant or exhibiting mycoparasitic activity, which involves the parasitism of one fungus by another [7,36]. Additionally, fungal endophytes frequently colonize host plants endophytically in a systemic manner to develop their resistance mechanisms towards fungal pathogens by inducing a systemic response [40]. However, fungal endophytes within a plant can have multifaceted effects on fungal pathogens depending on the specific fungal endophyte, host plant, and the pathogen species involved. The multifaceted interactions between fungal endophytes and plants may be antagonistic, directly or indirectly affecting the physiology and immune system of the host [59]. The reduction in dieback disease symptoms in *Fraxinus excelsior* seedlings by *Hypoxylon rubiginosum* was confirmed by producing antifungal metabolites, namely phomopsidin and its new derivative, 10-hydroxyphomopsidin, which served as the major secondary metabolites of *H. rubiginosum* that inhibit *Hymenoscyphus fraxineus* [41]. In in vitro dual-culture tests, endophytic isolates (such as *A. insulicola* and *A. melleus*) exhibited growth promotion in *Cucumis sativus* and strong antifungal activity against *Pythium aphanidermatum* with over a 50% suppression of growth by damaging the hyphal walls and abnormal mycelial growth with notably higher cellulase and β-glucanase activity [42]. Future research might employ fungal endophytes to improve agricultural plants’ ability to withstand infecting phytopathogens.

### 3.2. Role of Fungal Endophytes against Bacterial Pathogens

Bacterial diseases in various plant species can result in substantial economic losses due to reduced yields and disease management costs [60]. Some bacterial pathogens are highly destructive and possess the ability to destroy the entire crop. The management strategies used to combat bacterial plant diseases often involve disease-resistant plant varieties, crop rotation, and the application of copper-based bactericides and antibiotics [61]. It has been noted that using endophytes as biocontrol agents in sustainable agricultural practices can help reduce reliance on chemical treatments for bacterial diseases [51]. Additionally, fungal endophytes are known to improve crop health and safeguard plants from bacterial infections by reducing the prevalence of disease through the production of a broad spectrum of antibacterial secondary metabolites (viz., acetic acid, acetol, aliphatic compounds, alkaloids, hexanoic acid, peptides, phenylpropanoids, polyketides, and terpenoids) that target different bacterial pathogens [7].

The inoculation of root endophytic *Piriformospora indica* protects the root system, encourages growth, and confers disease resistance in *Anthurium andraeanum* plants by stimulating the host’s antioxidative enzyme activities and pathogenesis-related gene expression against *Ralstonia solanacearum* infection [52]. The potential endophytic *Fusarium solani* and *Trichoderma asperellum* were found to induce resistance against *R. solanacearum*, causing wilt disease in *Capsicum annuum* through changes in the activities of PAL, POD, β-1,3-glucanase, and total phenols, which are related to plant defense systems [53]. The lipid-rich endo-metabolites extracted from the vertically transmitted endophytic *Penicillium* sp. strain showed the potential to attenuate virulence factors of phytopathogenic *R. solanacearum*, even if the susceptible host-plant-associated endophytic fungus cannot directly combat *R. solanacearum* [62]. Recently, Huang et al. [54] noted that the secretion of the novel protein, Fusarium-lateritium-Secreted Protein (FlSp1), in endophytic *F. lateritium* could function as an immune-triggering effector to inhibit its colonization and enhance resistance against *R. solanacearum* in *Nicotiana benthamiana* through the upregulation of plant ROS along with plant immune system manipulation. The studies showed that fungal endophytes can serve as biocontrol agents and support plant defense against bacterial pathogens since they can produce antibacterial compounds (Table 1).

### 3.3. Role of Fungal Endophytes against Viral Pathogens

Plant viral pathogens were found to cause significant economic losses through reductions in crop yields and quality and can even kill infected plants [63], and hence, the management of these pathogens is necessary to protect crops and ensure food security. Management strategies for plant viral diseases include using virus-resistant crop varieties, rotating crops, managing vector hosts, avoiding infected plant materials, and practicing proper sanitation [64]. Some fungal endophytes that exhibit entomopathogenic activities against insect vectors may also indirectly reduce the transmission of viral diseases [55,56]. Fungal endophytes, viz., *M. anisopliae* and *T. harzianum*, were found to colonize the plants of *Z. mays* that help to reduce *Sugarcane mosaic virus* disease severity and virus titers, thereby conferring protective effects, as reported by Kiarie et al. [57]. Plant immune inducers, like ZhiNengCong, extracted from endophytic *Paecilomyces variotii*, exhibited high activity in enhancing *N. benthamiana* resistance at low concentrations to *Potato X virus* through the positive regulation of RNA silencing and salicylic acid signaling pathways [58]. Additionally, Lacerda et al. [65] reported that fungal endophytes are an abundant source of secondary metabolites with antiviral effects that combat viral diseases. Thus, it was noted that the endophytic fungal priming of plants offers a sustainable alternative method for managing viral diseases (Table 1).

### 3.4. Role of Fungal Endophytes on Pests

Plant pests, viz., mites, insects, etc., can cause serious damage to agricultural crops through their feeding activities on plant tissues and by transmitting plant viruses [66]. Effective pest management strategies, including integrated pest management and using pest-resistant crop varieties and chemical pesticides, are essential to mitigate the damage caused by pests and maintain crops’ overall health and yield [67]. Most terrestrial plants harbor many potential fungal endophytes, which modify plant defense mechanisms to directly or indirectly affect insect behavior and community structures [68]. Fungal endophytes substantially impact pest control and plant defense against many pest species [31,69,70,71]. Some fungal endophytes produce bioactive substances or secondary metabolites with insecticidal properties that are toxic or repellent to pests. The availability of resources that pests require for their survival and reproduction can be limited by fungal endophytes, which can compete for space and nutrients with plant pests, thereby indirectly reducing pest populations [31]. Fungal endophytes can also strengthen the host plant’s defense system against pests by activating plant metabolic pathways, which produce compounds that are toxic to pests [72].

Moreover, the endophytic fungal colonization of plants significantly affects the emission of pest-induced plant volatiles that may serve as pest repellents or natural enemy attractants, thereby directly controlling pests and boosting plant resistance against agronomically important pests [73]. Lana et al. [74] reported that *T. hamatum* was found to show entomopathogenic ability against the larvae of *Spodoptera littoralis* upon the application of its culture filtrates and spores through the large production of siderophore rhizoferrin, which was found to be responsible for the insecticidal activity. Recently, Darsouei et al. [75] provided further evidence that both the fungal endophytes, viz., *B. bassiana* and *B. varroae*, not only colonized the different parts (such as the leaves, stems, and roots) of *Beta vulgaris* plants upon seed inoculation or foliar spray without causing any apparent detrimental effects, but also enhanced plant growth and reduced the survival of *S. littoralis*. It was noted that using fungal endophytes in pest management strategies may offer eco-friendly and sustainable solutions to reduce pest damage and enhance crop protection (Figure 2) (Table 2).

### 3.5. Role of Fungal Endophytes in Plant-Parasitic Nematodes

Nematodes that parasitize plants threaten agriculture worldwide and have caused significant economic losses, particularly in tropical and subtropical areas [98]. It has been observed that nematodes can feed on various plant parts, viz., the roots, seeds, stems, leaves, and flowers. But most of their species primarily feed on plant roots and damage the root system by successfully establishing feeding sites and facilitating secondary pathogen entry. Plant-parasitic nematodes can act as vectors for several plant viruses, which can cause diseases and even plant death, escalating some of the agricultural challenges [99]. Conventional nematode control measures often involve the use of chemical nematicides. However, these chemicals may have unintended effects on ecosystems and beneficial soil microbial communities. There is considerable focus on exploring alternate methods for nematode management due to environmental and ecological concerns associated with chemical nematicides. Fungal endophytes have emerged as potential participants in the biocontrol of nematodes due to their capacity to prevent nematode growth and spread [47,100,101].

Fungal endophytes can mitigate plant-parasitic nematode infestation through different mechanisms, including parasitism, paralysis of the nematode, antibiosis, competition for space, and the secretion of lytic enzymes [101] (Figure 2). Fungal endophytes can produce nematocidal substances, parasitize nematode eggs and larvae, or physically trap nematodes and their eggs with hyphal loops [102,103,104]. Some endophytic fungi are known to secrete bioactive substances that directly or indirectly affect the nematode colonization of plants and the surrounding soil [105]. Similarly, fungal endophytes trigger induced systemic resistance and systemic acquired resistance, which facilitate chemical defense compound transport within the plant and promote the production of hormones (such as jasmonic acid, salicylic acid, and strigolactones) to develop resistance against nematodes [101]. Dutta et al. [106] have demonstrated that a gall-associated endophytic *A. niger* exhibited the complete mortality of *M. graminicola* juveniles in in vitro and in vivo conditions with ovicidal properties, reduced egg hatching, triggered *Oryza sativa* plant defense responses, and indirectly protected against *M. graminicola* infection. Hence, using fungal endophytes as a seed treatment or root inoculation is a promising strategy to control nematodes in agricultural plants. However, it is crucial to comprehend the bioactive substances that these endophytes produce to maximize their potential for nematode control and their potential as biocontrol agents against a broad range of nematodes (Table 3).

## 4. Role of Fungal Endophytes in Mitigating Abiotic Stress

Abiotic stresses (viz., drought, salt, heat, and heavy metals) are major environmental stresses that can substantially impact agricultural productivity and lead to global yield losses of over 50% [112]. These stresses can lead to irreversible damage, resulting in slow growth, serious injury, and even plant death [113]. Additionally, they cause growth and metabolism to be inhibited, which lowers the amount of biomass and active components in plants. Extensive studies in the literature have shown that fungal endophytes are essential for improving plants’ resilience to these abiotic stresses.

### 4.1. Drought Stress

Drought is a prominent abiotic stress that hampers the productivity of agricultural crops and poses direct climate challenges to achieve food security goals in developing countries [114,115]. Inadequate water availability has detrimental effects on plant life, encompassing osmotic stress induction, the blockage of photosynthesis, limited nutrient uptake, and over-production of ROS. In response to drought stress, plants manifest different biochemical, physiological, and molecular responses, viz., stomatal closure, reduced photosynthesis and transpiration rates, the production of stress hormones, the activation of the antioxidant defense system, and ROS production [116,117]. In this context, developing safe management strategies for sustainable crop production is essential to mitigate this stressful condition. Studies have evidenced that plant inoculation with fungal endophytes significantly reduces the negative impacts of drought stress with the positive nature of the plant–symbiont relationship, consequently demonstrating greater drought tolerance by improving the plant root biomass, total biomass, leaf morphology and anatomical structure, pigments, photosynthesis, stomatal conductance, transpiration, nutrient content, and antioxidant enzyme activities [118,119,120,121].

The mutualistic root endophytic *P. indica* was found to improve drought stress adaptation in *Hordeum vulgare* plants through colonizing plant roots, enhancing electron transfer chain and photosystem activity, and promoting protein accumulation, which is responsible for the primary metabolism, energy modulation, transporters, photorespiration, autophagy, and altering the host’s amino acid metabolism [122,123]. Miranda et al. [124] described that the endophytic strains of *Zopfiella erostrata* exhibited higher drought tolerance by extensively colonizing *S. lycopersicum* and *T. aestivum* plant roots, profusely forming melanized mycelium in the rhizosphere, enhancing plant biomass production, improving nutrient mineralization and water uptake, inducing proline accumulation, and decreasing the accumulation of lipid peroxide to control plants subjected to drought stress. The pronounced impact of fungal endophytes varies significantly based on specific endophytes, host plants, and water availability, thus indicating a certain level of compatibility between the plant and the strain. Therefore, fungal endophytes are essential for conferring plant resistance to drought stress (Table 4).

### 4.2. Salt Stress

Salt stress represents one of the significant abiotic stresses that negatively affects plant growth and productivity globally, especially in arid and semi-arid regions [163,164]. Prolonged exposure of plants to salt stress enhances intracellular osmotic pressure by producing an excessive amount of ROS. It causes the excess accumulation of sodium (Na^+^) to toxic levels in plants, thereby disturbing the K^+^/Na^+^ balance and plant metabolism and damaging the cellular membrane, chlorophyll, proteins, and nucleic acids [132,165]. Effective strategies to manage salt stress are crucial for promoting sustainable agriculture and facilitating optimal plant growth. Fungal endophytes induce plant salt tolerance by modulating endogenous hormones, defense gene expression, and the antioxidative system, which help reduce the ROS concentration [133,134] (Table 4). The mutualistic interaction of phytohormones secreting fungal endophytes could facilitate plant growth and counteract the negative impacts of salt stress by accumulating antioxidants and proline, regulating endogenous phytohormones, creating a physical barrier against salt uptake into the roots, and maintaining plant water potential and ionic homeostasis [135,136,137]. The root mutualistic endophytic *P. indica* had a noteworthy effect on *H. vulgare* biomass and growth and induced a systemic response by changing the proteome pattern and ion content in order to cope with salt stress [138,139]. Likewise, Jan et al. [132] have indicated that the endophytic relationship of *Yarrowia lipolytica* with *Z. mays* plants has notably enhanced plant growth in response to salt stress. This improvement is attributed to exogenous indole acetic acid secretion, the regulation of endogenous indole acetic acid and abscisic acid, and the production of CAT, POD, and proline in plants. Additionally, plant growth was increased when they were associated with the endophytic entomopathogenic *B. bassiana*, which alleviated the negative effects of salt stress, probably due to the accumulation of free proline and the activation of antioxidant enzymes in *Solanum tuberosum* plant tissues [140]. Future research might employ fungal endophytes to improve agricultural plants’ ability to withstand salt stress.

### 4.3. Heat Stress

Heat stress is one of the major abiotic stresses induced by high temperatures, which can have significant effects on plants by disrupting normal physiological functions such as enhanced protein denaturation and membrane fluidity, an elevated ROS content, the inactivation of chloroplast and mitochondrial enzyme activities, and decreased photosystem II-mediated electron transport [146]. Crop plants exhibit varied responses to heat stress during their life cycles, and these responses can be species-specific. The impact of heat stress on plants can be mediated by fungal endophytes, which adjust, regulate, or modify plants’ physiological, biochemical, and metabolic activities. The treatment of plants with fungal endophytes abolishes the negative impacts of heat stress via maintaining maximum photosystem II quantum efficiency, water use efficiency, photosynthesis, enhancing the root length and, inducing the accumulation of osmolyte and antioxidant enzyme activity relative to uninoculated plants [147,148,149]. The research clearly shows that the precise function of fungal endophytes has been investigated in relation to environmental acclimatization and the conferment of heat stress tolerance (Table 4).

### 4.4. Cold/Chilling Stress

Generally, cold stress induced by low temperatures can exacerbate the cell membrane lipid peroxidation through the excessive accumulation of malondialdehyde and hydrogen peroxide, consequently leading to plant death [166]. The colonization of fungal endophytes was found to reduce the contents of malondialdehyde and hydrogen peroxide. It alleviated the plant leaf cell damage caused by membrane lipid peroxidation under cold stress [51]. The root endophytic *P. indica* has also proven to be an effective agent to confer cold stress tolerance to host plants through mainly controlling cold-stress-related genes involved in metabolism pathways, phytohormone signaling, and lipid and inorganic ion transport [151]. Li et al. [152] have also concluded that endophytic *P. indica* confers *M. acuminata* seedlings with enhanced cold resistance through the stimulation of antioxidant capacity, soluble sugar accumulation, and the expression of cold-responsive genes (viz., *CSD1C*, *Why 1*, *HOS1*, and *CBF7-1*) in leaves. It was noted that fungal endophytes could be used to ameliorate cold stress tolerance in crop plants in the future.

### 4.5. Heavy Metal Stress

Heavy metal contamination seriously threatens aquatic life and ecosystems in developing countries, where industrial wastes and effluents are often directly discharged into rivers and accumulated on soil surfaces [167]. Agricultural areas may be adversely affected by the overuse of agricultural chemicals and the utilization of extensive municipal and industrial effluents for disposal in landfills. Heavy metals are toxic, non-biodegradable, persistent pollutants that can only transform from one chemical state to another, leading them to persist in the soil environment [168]. Heavy metals accumulate in plants via irrigated water and soil, impeding their development and transport to the food chain, and thus presenting possible health risks to animals and humans. Metals, namely cadmium, copper, zinc, cobalt, nickel, and lead, are toxic to many plants and other organisms when they are present in higher concentrations. Therefore, it is vital to implement effective measures to remove and remediate heavy metal pollution to safeguard human and ecological health.

In this context, fungal endophytes and plants have symbiotic relationships that are essential for enhancing the activity of antioxidant enzymes through heavy metal uptake from soil, followed by detoxifying the deleterious effects of heavy metal stress in plants [153,154,155]. It was observed that *P. indica* enhanced the resistance of *N. tabacum* plants to cadmium stress by upregulating the expression of photosynthesis-related proteins, GS, and POD to lessen the oxidative damage that cadmium causes [156]. Musa et al. [157] described that endophytic *Paecilomyces lilacinus* inoculation produces greater amounts of indole acetic acid and secondary metabolites that have an immense potential to promote *S. lycopersicum* plant growth and alleviate the adverse effects of heavy metal (such as cobalt and lead) stress conditions. Therefore, fungal endophytes play a role in heavy metal biosorption, thereby stabilizing the negative effects of heavy metal toxicity. This eco-friendly approach to the bioremediation of heavy metals helps eradicate heavy metal toxicity and fosters a healthy dietary ecosystem (Table 4).

## 5. Adverse Effects of the Use of Fungal Endophytes on Host Plants

Apart from the beneficial effects of fungal endophytes, some reported studies show that their adverse effects on the survival of host plants are minimal and thus could be ignored due to their lack of pathogenic properties. The result of the pathogenicity tests revealed that most of the fungal endophytes that reside inside host plant tissues can reflect their pathogenic ability with differing degrees of virulence to potentially infect both their natural hosts and non-hosts, including particularly important agricultural crop species [169,170,171]. Therefore, it was noticed that several fungal endophytes have also been reported to be part of a group of latent plant pathogens, which alter the host’s physiology and render them more prone to fungal infection.

## 6. Interaction between Fungal Endophytes and Other Members of Plant Microbiome

The interaction between fungal endophytes and other members of the plant microbiome is pivotal in influencing a plant’s response to environmental stress. The plant microbiome is a multifaceted community of microbes, encompassing bacteria, fungi, archaea, protists, nematodes, viruses, and other microbes that colonize and inhabit plant tissues and their surroundings [172]. Fungal endophytes frequently establish synergistic relationships with plant-growth-promoting bacteria (PGPB) that improve plant growth and development through increasing nutrient availability, producing plant growth hormones, and suppressing pathogenic microbes [173]. The combined effects of fungal endophytes and PGPB can enhance a plant’s tolerance to environmental stresses, such as drought, salinity, and nutrient deficiency. Fungal endophytes may compete with other plant microbiomes for nutrients and space. Some fungal endophytes also produce diverse secondary metabolites with antimicrobial properties, which may act antagonistically against potential pathogens or other nearby microbes. As mycorrhizal fungi establish symbiotic associations with plant roots, the interaction between fungal endophytes and arbuscular mycorrhizal fungi can significantly impact nutrient cycling and transfer within the plant [174]. These interactions can influence the plant’s capacity to acquire nutrients and respond to environmental stress efficiently.

## 7. Challenges and Future Aspects

Researchers are now focusing on alternative eco-friendly solutions for sustainable food production as a countermeasure to the increasing challenges of pollution, the uncontrolled usage of agrochemicals, population explosion, food imbalance, and depleting fertile land. Fungal endophytes have garnered much attention in the agricultural field because they reduce biotic and abiotic stresses and foster mutualistic interactions that enhance plant development. Identifying novel fungal endophytes signifies diverse host associations that can flourish in harsh environmental conditions. Utilizing fungal endophytes as biofertilizers independently or in groups as part of integrated biotic and abiotic stress management strategies could offer a remedial alternative to decrease reliance on agrochemical applications. Establishing consortia with potential fungal endophyte strains from neighboring agricultural farms that produce sophisticated biocontrol products can be a crucial and environmentally friendly measure towards sustainability to benefit both farmers and the ecosystem. Comprehensively studying the endophytic fungal community by employing recent advanced biotechnological and bioinformatic tools, viz., RNA interference, the CRISPR–Cas system, metabolomics, and the next-generation sequencing system, can unveil the intricate interaction between fungal endophytes and hosts, as well as the molecular patterns involved in stress tolerance mechanisms. Advanced investigations into specific host fungal endophytes are essential to develop robust bio-inocula that will enable the production of organic food crops and make the future safer.

Incorporating these endophytes into cropping systems via foliar applications, seed treatments, or other methods can enhance agricultural efficiency and yield positive environmental effects concurrently. The main challenge in understanding the adverse effects of fungal endophytes on host plants is the potential toxin transmission into the food chain. Additionally, investigating whether fungal endophytes act as latent pathogens when artificially inoculated on another crop plant is crucial. It is important to consider that some endophytic fungi that are of importance for a particular species (host) may be pathogenic to other plants (non-host), with differing degrees of virulence that directly depend on the physiological/anatomical features of the plant species and signal transduction during the colonization process. It may also be noted that these beneficial endophytic fungi may transform into latent pathogens due to changes at the morphological, physiological, biochemical, and molecular levels within the plants during their growth cycles and ever-changing environmental factors. Additionally, reports also support the notion that when conditions become favorable, fungal endophytes transform into pathogenic forms, potentially infecting the host and other plants. The next challenge is comprehending how newly incorporated endophytes interact with the endobiomes of native plants. Its significance may also be recognized because fungal endophytes only produce antimicrobials when other endophytes are present, and not in a pure culture. On the contrary, some incorporated endophytes may eradicate natural endophytes during colonization or induce disease in new host plants. Therefore, researchers must thoroughly examine fungal endophytes and their potential pathogenic effects before their incorporation as biocontrol agents or biofertilizers.

Further research is necessary due to the lack of substantial information for the field performance of incorporated endophytes and potential alterations in their functioning caused by climatic changes. The potential challenges that limit the large-scale commercial incorporation of fungal endophytes include a lack of knowledge on specific plant–endophyte interactions, pathogen–plant defense systems, physio-chemical barrier, gene expression, and defense signaling pathways in agriculture. Finally, registering bio-formulated or bioagent products of fungal endophytes prior to their commercialization is a daunting challenge as it takes a lot of time before their release in the environment is approved due to licensing rules and regulatory organizations. Renowned scientists are contributing and hopefully succeeding in developing novel commercially feasible fungal endophyte formulations and inoculations, ultimately enabling sustainable and climate-smart agricultural practices.

## 8. Conclusions

In conclusion, harnessing the potential of fungal endophytes offers a sustainable and effective approach to reducing crop losses while minimizing the adverse effects of conventional chemical methods. This strategy addresses the challenges caused by biotic and abiotic stresses in plants and contributes towards the reduced usage of chemical pesticides and fertilizers in agriculture. Fungal endophytes represent valuable biological tools in developing environmentally sustainable and economically viable agricultural practices because they promote soil health, enhance plant resilience, and ensure long-term food security.

## Figures and Tables

**Figure 1 jof-10-00116-f001:**
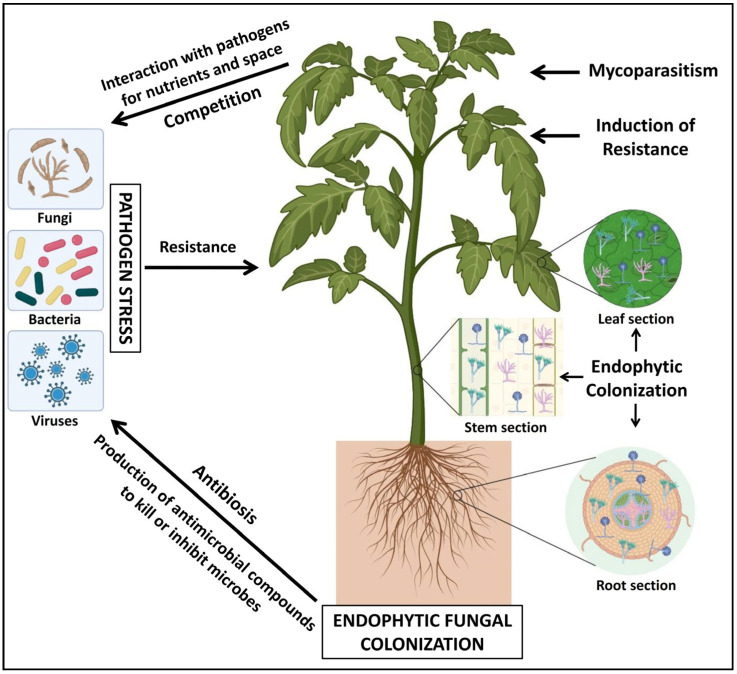
Mechanism of action of fungal endophytes in the biocontrol of pathogens in host plants.

**Figure 2 jof-10-00116-f002:**
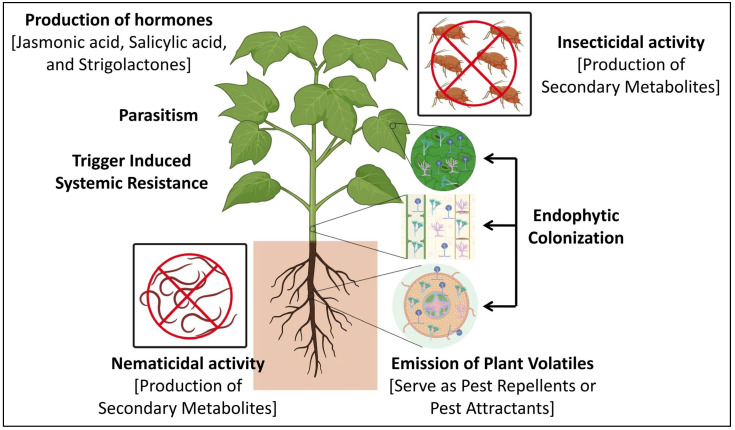
Mechanism of action of fungal endophytes in biocontrolling pests and nematodes in host plants.

**Table 1 jof-10-00116-t001:** Fungal endophytes and their roles in biocontrolling or antagonizing pathogens in host plants.

Endophytic Fungi	Host	Role in Pathogen Control	Role in Host Plant	Reference
**Fungal pathogens**				
*B. bassiana*	*S. lycopersicum*	Biocontrol of *A. alternata* and *B. cinerea*	Colonized the plants, improved plant growth, and inhibited disease development	[31]
*Aspergillus calidoustous*, *Diaporthe phaseolorum*, *P. citrinum* and *T. asperellum*	*Elaeisguineensis*	Antagonistic nature towards *Ganoderma boninense*	Showed the most rapid colonization and compatibility, produced volatiles and non-volatiles along with competitive exclusion	[35]
*M. brunneum* and *Beauveria bassiana*	*Capsicum annuum*	Biocontrol of *F. culmorum*, *F. moniliforme*, and *F. oxysporum*	Inhibited *Fusarium* spp. growth, showed competition for resources or niche, antibiosis, colonized plants, reduced crown and root rot disease severity and incidence with improved plant growth	[36]
*Serendipita indica*	*Musa acuminata*	Biocontrol of *F. oxysporum*	Improved the plant resistance to *F. oxysporum* and increased the activities of ascorbate, CAT, GR, SOD, and POD enzymes	[37]
*A. terreus*	*Zingiber officinale*	Biocontrol of *Colletotrichum gloeosporioides*	Colonized the same ecological niche and produced a bioactive metabolite, terrein, against *C. gloeosporioides*	[39]
*B. bassiana*	*Vitis vinifera*	Protective potential against *Plasmoparaviticola*	Achieved the highest plant colonization percentage, significantly reduced downy mildew disease severity, and upregulated diverse defense-related genes in plants	[40]
*H. rubiginosum*	*F. excelsior*	Biological control of *H. fraxineus*	Inhibited *H. fraxineus*, reduced dieback disease symptoms in seedlings, produced antifungal metabolites such as phomopsidin and 10-hydroxyphomopsidin	[41]
*A. insulicola* and *A. melleus*	*C. sativus*	Antagonistic activity against *P. aphanidermatum*	Suppressed *P. aphanidermatum* growth through damage of hyphal wall, increased electrolyte leakage, produced cellulase and β-glucanase enzymes, increased plant shoot length and dry mass	[42]
*Metarhiziumrobertsii*	*Phaseolus vulgaris*	Biocontrol of *F. solani*	Colonized the plants, showed antagonism, inhibited conidial germination and growth, produced heat-stable inhibitory metabolite, showed lower disease indices and better plant growth	[43]
*Acremonium* sp., *Leptosphaeria* sp., *P. simplicissimum* and *Talaromyces flavus*	*Gossypium hirsutum*	Biocontrol of against *Verticillium dahliae*	Decreased Verticillium wilt disease index and incidence, improved cotton bolls and cotton yield, increased transcript levels for POD, PPO, and PAL	[44]
*M. brunneum* and *B. bassiana*	*Triticum aestivum*	Control of *F. culmorum* infection	Systemically colonized plant roots and shoots, promoted plant growth parameters, significantly reduced crown and root rot disease severity and incidence	[45]
*Purpureocillium lilacinum*	*P. vulgaris*	Biocontrol of *Sclerotinia sclerotiorum*	Significantly reduced *S. sclerotiorum* disease severity through prevention of sclerotia formation, mycelial growth, and myceliogenic and carpogenic germination; increased cell membrane permeability and lipid peroxidation of *S. sclerotiorum* mycelia; decreased oxalic acid; and improved POD, PPO, and PAL activity in plants	[46]
*P. brefeldianum*	*Cucumis melo*	Biocontrol of *F. oxysporum*	Showed antifungal effects and reduced *Fusarium* wilt disease severity, produced major bioactive compound brefeldin A, and dramatically increased the population of *P. brefeldianum* in plants	[47]
*B. bassiana*	*C. annuum* and *Solanum lycopersicum*	Antagonistic activity against *Botrytis cinerea*	Internally colonized different plant parts and showed antagonism	[48]
*T. asperellum*, *T. hamatum*, and *T. harzianum*	Sweet corn	Biocontrol of *Exserohilum turcicum*	Controlled northern corn leaf blight disease by showing potent competitive and effective antifungal activity	[49]
*T. asperellum*	*Z. mays*	Biocontrol of *E. turcicum*	Encircled *E. turcicum* hyphae effectively, reinforced antagonistic behavior, enhanced seed germination, improved plant growth, and suppressed *E. turcicum* infection	[50]
**Bacterial pathogens**				
*Acrocalymma* sp., *Fusarium* sp., *Curvularia* sp., *Phialocephala*, *Setophoma*/*Edenia* and *Trichoderma* sp.	*Brassica oleracea*	Powerful defensive capacity against *Xanthomonas campestris*	Decreased the damage caused by pathogenic bacteria, reduced the disease incidence, and activated plant systemic resistance against *X. campestris*	[51]
*P. indica*	*A. andraeanum*	Strong potential against *R. solanacearum*	Extensively colonized the plant roots, shortened *Anthurium* recovery period, promoted plant growth, conferred disease resistance, induced faster elongation of *Anthurium* roots, exhibited higher photosynthesis rate, and increased phosphate absorption, activities of antioxidative enzymes, and relative expressions of *ERF*, *LOX*, *VSP*, *NPR1*, *PR1*, and *PR5*	[52]
*F. solani* and *T. asperellum*	*C. annuum*	Biocontrol of *R. solanacearum*	Reduced bacterial wilt development, increased crop yield, and enhanced enzyme activities (POD, β-1,3-glucanase, PAL and PPO) and total phenols	[53]
*F. lateritium*	*N. benthamiana*	Conferred resistance to *R. solanacearum*	Secreted novel protein [Fusarium-lateritium-Secreted-Protein (FlSp1)], reduced the fungal colonization, enhanced plant resistance, and regulated plant ROS burst and immune system	[54]
**Viral pathogens**				
*B. bassiana*	*Cucurbita pepo*	Conferred protection against *Zucchini yellow mosaic virus*	Successfully colonized plants, significantly lowered disease incidence and severity	[55]
*Hypocrea lixii*	*Allium cepa*	Biocontrol of *Iris yellow spot virus*	Endophytically colonized plants, significantly lowered disease level, reduced replication of *Iris yellow spot virus*	[56]
*M. anisopliae* and *T. harzianum*	*Z. mays*	Protective role against *Sugarcane mosaic virus*	Colonized plant tissues, reduced *Sugarcane mosaic virus* disease severity and virus titer levels	[57]
*P. variotii*	*N. benthamiana*	Resistance to *Potato X virus*	Exhibited the antiviral activity of plant immune inducers like ZhiNengCong on plant, induced ROS accumulation, increased salicylic acid content, upregulated PAL gene expression, activated salicylic acid signaling pathway, and promoted RNA silencing	[58]

Note: PAL—phenylalanine ammonia lyase; PPO—polyphenol oxidase; POD—peroxidase; GR—plutathione reductase; CAT—catalase; SOD—superoxide dismutase; *ERF*—ethylene responsive factor; *LOX*—lipoxygenase; *VSP*—vegetative storage protein; *NPR1*—non-expresser of pathogenesis-related; *PR1*—pathogenesis-related 1; *PR5*—pathogenesis-related 5.

**Table 2 jof-10-00116-t002:** Fungal endophytes and their roles in biocontrolling pests in host plants.

Fungal Endophyte	Host Plant	Role in Pest Control	Role in Host Plant	Reference
*B. bassiana*	*S. lycopersicum*	Control of *Macrosiphum euphorbiae*	Colonized the plants, promoted plant growth, and significantly reduced survival and fertility of *M. euphorbiae*	[31]
*Fusarium* sp., *Setophoma*/*Edenia* and *Curvularia* sp.	*B. oleracea*	Conferred resistance towards *Mamestrabrassicae* larvae	Activated plants’ systemic resistance against *M. brassicae* through decrease in damage index as noted through decreased leaf area consumption by the larvae	[51]
*Hypocrea lixi*	*A. cepa*	Control of *T. tabaci*	Colonized the plants and significantly lowered the number of feeding punctures	[56]
*B. bassiana*, *Gibberella moniliformis*, *H. lixi*, *M. anisioplaie* and *T. asperellum*	*Vicia faba*	Control of *Aphis fabae* and *Acyrthosiphonpisum*	Significantly lowered nymph number in *A. fabae* and *A. pisum*, exhibited detrimental effect on offspring fitness, fecundity, and development along with enhanced seedling survivorship	[68]
*B. bassiana*	*V. vinifera*	Control of *Empoascavitis* and *Planococcusficus*	Reduced infestation rate and growth of *E. vitis* and *P. ficus*	[69]
*B. bassiana*	*Z. mays*	Control of *Sitobion avenae* population	Colonized the plant significantly and reduced the survival and fecundity of *S. avenae*	[70]
*B. bassiana*	*Carya illinoinensis*	Control of *Galleria mellonella*, *Tenebrio molitor*, *Curculio caryae*, *Melanocallis caryaefoliae*, and *Monellia caryella*	Colonized seedlings, established in different plant parts, retained pathogenicity against *G. mellonella*, *T. molitor*, and *C. caryae* and significantly reduced the population of both *M. caryaefoliae* and *M. caryella*	[71]
*B. bassiana*	*S. lycopersicum*	Conferred resistance against *B. tabaci*	Inhibited the reproduction of *B. tabaci*, stimulated plant defenses and induced systemic resistance, and activated metabolic pathways (viz., tryptophan, flavonoids, and alkaloids) in plants	[72]
*B. bassiana* and *B. varroae*	*B. vulgaris*	Control of *S. littoralis*	Colonization rate increased over the time, which helped in the enhancement of plant growth and reduced *S. littoralis* larval weight gain, decreased lipase and protease activity in *S. littoralis* gut, and reduced survival of *S. littoralis* pupae and eggs laid by female moths	[75]
*M. anisopliae*	*Brassica napus*	Control of *Plutella xylostella* larvae	Colonized the internal tissues of plants and showed significant differences in the mean % *P. xylostella* larval mortality	[76]
*B. bassiana*	*Corchorus capsularis*	Control of *Apioncorchori*	Colonized plant leaves showed the highest colonization frequency and reduced *A. corchori* infestation	[77]
*P. lilacinum* and *B. bassiana*	*G. hirsutum*	Control of *A. gossypii*	Colonized the plant, negatively affected the reproduction of *A. gossypii*, and significantly lowered the number of *A. gossypii* on plants	[78]
*H. lixii*, *Clonostachys rosea*, *Fusarium* sp., *T. asperellum*, *T. harzianum* and *T. atroviride*	*A. cepa*	Effects on *Thrips tabaci*	Colonized the plants effectively with higher mean percentage recovery, significantly lowered the number of feeding punctures and eggs laid by adult *T. tabaci*	[79]
*B. bassiana* and *P. lilacinum*	*G. hirsutum*	Control of *Helicoverpa zea* larvae	Colonized plants, enhanced plant growth, and reduced the survival and development of *H. zea* larvae	[80]
*B. bassiana*, *Isaria fumosorosea*, and *M. robertsii*	*Sorghum bicolor*	Control of of *Sesamia nonagrioides* larvae	Prevented *S. nonagrioides* larvae from entering stalks, reduced larval mortality and tunnel lengths, and protected plants from damage	[81]
*B. bassiana*	*S. lycopersicum*	Control of *Helicoverpa armigera*	Colonized the seedlings and achieved the highest larval mortality of *H. armigera* and reduced the effect of *H. armigera*	[82]
*B. bassiana* and *H. lixii*	*P. vulgaris*	Control of *Liriomyza* leafminer flies (like *L. huidobrensis*, *L. trifolii*, and *L. sativae*)	Colonized different parts of the plant and showed lower leafminer infestation, varied mean pupae number from infested leaves, and higher seed yield	[83]
*M. anisopliae*		Control of *Ophiomyia phaseoli*	Colonized different plant parts, significantly reduced the feeding and oviposition and number of pupae and adult emergence of *O. phaseoli*	[84]
*B. bassiana* and *M. brunneum*	*C. melo*, *Lycopersicon esculentum* and *Medicago sativa*	Control of *S. littoralis* larvae	Colonized the plant and offered a high *S. littoralis* larval mortality rate	[85]
*Chaetomium globosum*	*G. hirsutum*	Control of *A. gossypii* and *S. exigua*	Negatively affected the reproduction, development, and fecundity of both cotton *A. gossypii* and *S. exigua*	[86]
*Lecanicillium lecanii*, *I. fumosorosea* and *B. bassiana*	*P. vulgaris*	Control of *Tetranychus urticae*	Colonized the plant; increased plant height and fresh weight; and reduced larval survivorship, development, adult longevity, female fecundity, and reproduction of *T. urticae*	[87]
*M. brunneum* and *B. bassiana*	*Capsicum annum*	Control of *Aphidius colemani* and *Myzus persicae*	Colonized different plant parts, enhanced several plant growth parameters, and controlled development, fecundity, and reproduction of *A. colemani* and *M. persicae*	[88]
*B. bassiana*	*Glycine max*	Control of *Helicoverpa gelotopoeon*	Protected plants against *H. gelotopoeon*; significantly reduced mean duration of larval stages, adult stages, and total life cycle duration; reduced oviposition period, fertility, and fecundity of *H. gelotopoeon*; and reduced leaf consumption by *H. gelotopoeon*	[89]
	*T. aestivum and T. durum*	Control of *S. littoralis* larvae	Successfully established within and colonized the plants, boosted spike production in plants, and increased grain yield and plant root length with significant higher mortality in *S. littoralis* larvae	[90]
	*Citrus limon*	Control of *Diaphorina citri*	Successfully colonized the seedlings; improved plant height and flush production; caused adult mortality and egg production; and reduced *D. citri* adult emergence	[91]
*M. brunneum* and *B. bassiana*	*C. melo*	Control of *A. gossypii*	Colonized the plants, offered higher mortality and fecundity on *A. gossypii*	[92]
*M. anisopliae*, *I. fumosorosea*, and *B. bassiana*	*C. annum*	Control of *M. persicae*	Affected mortality and population of *M. persicae* in planta, caused feeding disorders and disrupted reproduction cycle	[93]
*B. bassiana*	*Z. mays*	Control of *Rachiplusia nu*	Colonized plants; increased percentages of seed germination, plant height, leaf number, grain weight, and yield; and significantly affected leaf area consumed by *R. nu* larvae	[94]
*M. robertsii*, *I. fumosorosea*, and *B. bassiana*	*S. bicolor*	Control of *S. nonagrioides* larvae	Induced *S. nonagrioides* larval mortality and decreased their relative growth rate, infestation, and tunneling length; showed relatively higher virulence; decreased food consumption and feces produced by *S. nonagrioides* larvae; and slightly changed the digestibility	[95]
*B. bassiana*	*S. lycopersicum*	Conferred resistance against *Bemisia tabaci*	Effectively colonized plants, uniformly distributed among plant parts, and promoted plant growth	[96]
*B. bassiana*	*B. oleracea*	Control of *P. xylostella* and *M. persicae*	Highly colonized sites of fungal exposure inside plants, showed maximum mortality of *P. xylostella* and *M. persicae*	[97]

**Table 3 jof-10-00116-t003:** Fungal endophytes and their roles in biocontrolling nematodes in host plants.

Fungal Endophyte	Host Plant	Nematode Control	Role in Host Plant	Reference
*P. brefeldianum*	*C. melo*	Biocontrol of *M. incognita*	Showed anti-nematodal activity, significantly reduced the gall numbers, produced the major bioactive compound brefeldin A, dramatically increased the population of *P. brefeldianum* on plants, and caused higher accumulation of brefeldin A in plant roots	[47]
*C. globosum*	*G. hirsutum*	Biocontrol of *M. incognita*	Inhibited *M. incognita* infection and reduced female reproduction	[86]
*Phialemonium inflatum*		Biocontrol of *M. incognita*	Reduced root penetration by juvenile *M. incognita*, significantly suppressed *M. incognita* galling of roots and egg production and improved plant growth	[100]
*Fusarium* spp., *Chaetomium* sp., *Acremonium* sp., *Trichoderma* sp., *Phyllosticta* sp., and *Paecilomyces* sp.	*C. sativus*	Biocontrol of *M. incognita*	Decreased gall number, produced nematodes and compounds to affect the motility of second stage of *M. incognita* juveniles, highly colonized roots and aboveground parts of seedlings	[102]
*Acremonium implicatum*	*Lycopersicon eseulentum*	Biocontrol potential of *M. incognita*	Inhibited second stage of *M. incognita* juveniles, suppressed egg hatching, inhibited root gall formation, reduced *M. incognita* population in soil, and showed lower root gall index of plants	[103]
*T. asperellum*, *F. solani* and *F. oxysporum*	*S. lycopersicum*	Biocontrol of *M. incognita*	Reduced penetration, galling, and reproduction of *M. incognita* and decreased egg density of *M. incognita*	[104]
*F. moniliforme*	*O. sativa*	Antagonistic activity against *M. graminicola*	Decreased *M. graminicola* penetration into plant roots and enhanced male-to-female ratio, reduced *M. graminicola* invasion, and showed repellent effect on nematode movement	[105]
*A. niger*	*O. sativa*	Biocontrol of *M. graminicola*	Exhibited 100% juvenile mortality of *M. graminicola*; showed ovicidal property; reduced egg hatching; significantly showed lower number of *M. graminicola* juveniles; decreased root galling index, number of juveniles penetrating the root, and reproduction; triggered plant defense responses; and indirectly provided protection against *M. graminicola* infection	[106]
*F. oxysporum*	*C. pepo* and *C. melo*	Biocontrol of *Meloidogyne incognita*	Decreased early plant root penetration of *M. incognita*	[107]
*P. indica*	*Arabidopsis thaliana*	Antagonistic potential against *Heterodera schachtii*	Colonized plant roots and significantly affected the vitality, infectivity, reproduction and development of *H. schachtii*	[108]
*F. oxysporum* and *Rhizobium etli*	*S. lycopersicum*	Biocontrol of *M. incognita*	Enhanced plant resistance toward *M. incognita*, decreased number of eggs and juveniles of *M. incognita*, and reduced root penetration, reproduction, and development of *M. incognita*	[109]
*P. indica*	*G. max*	Biocontrol oft *H. glycines*	Significantly decreased egg population density of *H. glycines*, showed strong growth- and yield-promoting effects on *G. max*, increased shoot biomass, accelerated plant development, and increased flowering	[110]
*F. oxysporum*	*A. thaliana*	Biocontrol of *M. incognita*	Colonized plant roots without causing disease symptoms, systemically reduced *M. incognita* infection development and fecundity, promoted plant growth, and significantly decreased number of *M. incognita* juveniles and galls produced	[111]

**Table 4 jof-10-00116-t004:** Fungal endophytes and their roles in mitigating abiotic stress in host plants.

Endophytic Fungi	Host	Role in Host Plant	Reference
**Drought stress**			
*P. minioluteum*	*Chenopodium quinoa*	Colonized plants, affected growth of radicles, improved the formation of roots, and increased plant resistance and positive nature of plant–symbiont interaction	[118]
*Darksidea* strain, *Knufia* sp., and *Leptosphaeria* sp.	*Ammopiptanthus mongolicus*	The endophyte formed a strain-dependent symbiotic relationship with plants and increased the total plant biomass	[119]
*Embellisia chlamydospora*, *Knufia* sp., *Leptosphaeria* sp., and *Phialophora* sp.	*Hedysarum scoparium*	Successfully colonized plant roots, established a positive symbiosis with host plants, and increased total plant biomass, antioxidant enzyme activities, and nutrient content	[120]
*Acrocalymma vagum*	*Ormosia hosiei*	Enhanced leaf morphology and anatomical structure, stomatal conductance, transpiration rate, net photosynthetic rate, and pigment content; lowered the intracellular CO_2_ concentration; and preserved mitochondria, chloroplasts, and cell membrane	[121]
*P. indica*	*H. vulgare*	Colonized and increased the plant biomass and accumulated proteins involved in ROS scavenging, photosynthesis, plant defense responses, and signal transduction	[122]
*P. indica*	*H. vulgare*	Colonized plant roots; increased activity of electron transfer chain and photosystem; accumulated proteins responsible for primary metabolism, energy modulation, photorespiration, autophagy, and transporters; and altered host’s amino acid metabolism	[123]
*Z. erostrata*	*S. lycopersicum* and *T. aestivum*	Profusely formed melanized mycelium in rhizosphere, exhibited higher tolerance to drought, improved nutrient mineralization and water uptake, enhanced plant biomass production, induced accumulation of proline, and decreased lipid peroxide accumulation	[124]
*Neotyphodium coenophialum*	*Lolium arundinaceum*	Caused significantly greater tillering and survival of re-watered plants, higher levels of free fructose, glucose, trehalose, glutamic acid, proline, and sugar alcohols in plants and increased fungal metabolites such as alkaloids, mannitol, and loline	[125]
Ascomycota sp. and *Cladosporium cladosporioides*	*N. benthamiana*	Colonized and enhanced plant tolerance; delayed wilting of shoot tips; increased relative water content, plant biomass, proline, soluble protein, soluble sugar, and activity of antioxidant enzymes (such as PPO, POD, and CAT); reduced ROS production and electrical conductivity; and upregulated drought-defense-related genes	[126]
*Nectria haematococca*	*S. lycopersicum*	Significantly improved plant growth parameters, induced drought stress tolerance, and significantly enhanced proline accumulation in shoots	[127]
*A. vagum*, *F. acuminatum* and *Paraboeremia putaminum*	*Glycyrrhiza uralensis*	Colonized and formed strain-dependent symbiosis with plants; increased plant biomass and glycyrrhizin content; improved plant root development, nutrient absorption, photosynthetic and antioxidant parameters; and altered the soil microbiota	[128]
*A. chlamydospora* and *Preussia terricola*	*G. uralensis*	Colonized the plant roots, increased the total plant biomass and root biomass; and caused higher available nitrogen, soil organic matter, and glycyrrhizic acid contents	[129]
*Neocamarosporium* sp. and *Periconia macrospinosa*	*C. sativus* and *S. lycopersicum*	Improved plant growth, chlorophyll, proline content, and antioxidant enzymatic activities	[130]
*A. aculeatus*, *Meyerozyma guilliermondi* and *Microdochium majus*	*Moringa oleifera*	Improved plant growth attributes, total chlorophyll, carotenoids, and primary and secondary metabolites; decreased abscisic acid level; increased activity of antioxidant enzymes, viz., CAT, APX, and total antioxidant capacity; reduced ROS production; caused larger stomatal aperture and lesser decrease in water potential; and upregulated *MolAPX*, *MolHSF3*, and *MolHSF19* gene expression	[131]
**Salt stress**			
*Neocamarosporium* sp. and *P. macrospinosa*	*C. sativus* and *S. lycopersicum*	Enhanced plant growth, chlorophyll, proline, and antioxidant enzymatic activity	[130]
*Y. lipolytica*	*Z. mays*	Significantly promoted plant growth attributes, like higher chlorophyll and carotenoid contents, reduced electrolyte leakage, higher relative water content of seedlings, lower endogenous abscisic acid, and higher endogenous indole acetic acid, and significantly controlled production of proline, CAT, and POD	[132]
*Bipolaris* sp.	*G. max*	Produced organic acids, like indole acetic acid and gibberellins; showed salt stress resistance; enhanced plant length, weight, and chlorophyll; increased salicylic acid; decreased endogenous abscisic acid; caused higher level of antioxidants and oxidative stress markers, viz., PPO, POD, superoxide anion, and malondialdehyde; improved plant resistance to NaCl stress; and decreased *GmFDL19*, *GmNARK*, and *GmSIN1* expression levels	[133]
*A. terreus*	*O. sativa* and *Z. mays*	Substantially increased plant biomass, relative water content, photochemical efficiency, and oxidative balance; enhanced gibberellic acid concentration; upregulated photosynthesis and antioxidant defense cascade; downregulated oxidative damage markers, like hydrogen peroxide and malondialdehyde; and displayed positive plant–microbe interaction	[134]
*Paecilomyces formosus*	*C. sativus*	Produced indole acetic acid and gibberellins, enhanced plant shoot length and allied growth characteristics, counteracted negative impacts of salt stress, accumulated antioxidants and proline, maintained water potential, reduced membrane damage and electrolytic leakage, and lowered the levels of endogenous abscisic acid content	[135]
*F. verticillioides*	*G. max*	Caused higher germination of seeds and plant growth; produced gibberellins; significantly enhanced plant length and fresh weight; effectively lessened negative effects of salt stress; decreased lipid peroxidation; enhanced protein content and activity of antioxidant enzymes, viz., POD, CAT, and SOD; and showed lower abscisic acid and elevated salicylic acid contents	[136]
*Stemphylium lycopersici*	*Z. mays*	Promoted activity of antioxidant enzymes (viz., APX and CAT), indole acetic acid content, phenolics and flavonoids, decreased malondialdehyde content, Na^+^ and Cl^−^ ion content, Na^+^/K^+^ and Na^+^/Ca^2+^ ratios, and increased Mg^2+^, K^+^, Ca^2+^, P, and N contents	[137]
*P. indica*	*H. vulgare*	Significantly enhanced plant growth and shoot biomass, modulated ion accumulation, increased foliar potassium/sodium ratio, and accumulated proteins associated with signal transduction, energy production, protein translation and degradation, photosynthesis, cell wall arrangement, and antioxidant defense	[138]
*P. indica*	*H. vulgare*	Helped in the identification of differentially regulated genes, metabolites, and ions to infer stress tolerance	[139]
*B. bassiana*	*S. tuberosum*	Improved plant growth, diminished adverse impact of salt stress, enhanced activity of antioxidant enzymes (such as SOD and POD), accumulated free proline, and increased stolon number	[140]
*Sordariomycetes* sp. and *Melanconiella elegans*	*Vigna unguiculata*	Improved colonization rate, plant growth attributes, stomatal conductance, photosynthesis, transpiration, and mineral nutrition	[141]
*P. chrysogenum* and *P. brevicompactum*	*Lactuca sativa* and *S. lycopersicum*	Greater biomass production, developed survival rate, diminished salt stress effects, maintained ionic homeostasis, enhanced *NHX1* gene expression, provoked increased photosynthetic energy generation efficiency, increased Na^+^ sequestration in vacuoles, and upregulated vacuolar *NHX1* Na^+^/H^+^ antiporter expression	[142]
*A. chlamydospora*, *Chaetomium coarctatum* and *F. equiseti*	*T. aestivum*	Improved plant seedling emergence and root growth, and exhibited the highest leaf sugar and proline contents	[143]
*C. globosum* and *Microsphaeropsis arundinis*	*T. aestivum*	Successfully colonized plant, promoted plant growth, and caused higher seed germination rate and biomass	[144]
*F. clavum*	*C. melo*	Exhibited plant-growth-promoting activities, viz., production of indole acetic acid and hydrolytic enzymes and phosphate solubilization; penetrated plant root tissues; improved plant height, weight, leaf number, stomatal conductance, photosynthesis, transpiration, membrane stability, and electrical conductivity; improved K^+^ absorption; reduced Na^+^ and Cl^–^ ion absorption; improved CAT, SOD, GPX, phenolic content, and chlorophyll content; decreased lipid peroxidation; increased proline accumulation; reduced superoxide ion production, hydrogen peroxide level, and cell mortality; and enhanced lignin deposition	[145]
**Heat stress**			
*A. flavus*	*H. annuus* and *G. max*	Produced secondary metabolites; caused higher salicylic acid, indole acetic acid, phenolic, and flavonoid contents; higher levels of plant abscisic acid and proline; and lower levels of flavonoids, phenols, AAO, and CAT in plants	[146]
*Thermomyces* sp.	*C. sativus*	Eliminated the negative effect of heat stress; maintained maximum photosystem II quantum efficiency, water use efficiency, and photosynthesis; enhanced root length; and accumulated saponins, flavonoids, total sugars, soluble proteins, and antioxidant enzyme activity	[147]
*Thermomyces lanuginosus*	*Cullen plicata*	Showed effective plant-growth-promoting activity, enhanced plant survival capacity, and increased total carbohydrate, flavonoid, and ascorbic acid contents and level of antioxidant enzymes (viz., PAL, POD, and CAT)	[148]
*A. niger*	*G. max* and *H. annuus*	Boosted plant biomass, height, and chlorophyll; curtailed ROS concentration and lipid peroxidation; augmented ROS scavenging activity, such as GR, CAT, AAO, POD, and SOD; enhanced phenolics and proline; and reduced abscisic acid concentration	[149]
*A. japonicus*	*G. max* and *H. annuus*	Displayed higher concentrations of indole acetic acid, salicylic acid, phenolics, and flavonoids; improved plant biomass; mitigated heat stress effects; negotiated activities of CAT, AAO, and abscisic acid; and improved nutritional quality (viz., phenolics, flavonoids, lipids, proteins, and soluble sugars) of seedlings	[150]
**Cold stress**			
*Fusarium* sp. and *Pyrenophora* sp.	*B. oleracea*	Promoted plant growth and increased cold tolerance	[51]
*P. indica*	*A. thaliana*	Upregulated cold stress response genes, viz., *WRKY*, *ERF*, *bHLH*, *HSF*, *MYB*, and *NAC* transcription factors	[151]
*P. indica*	*M. acuminata*	Reduced content of malondialdehyde and hydrogen peroxide; increased activities of SOD and CAT and contents of soluble sugar and proline; declined maximum photochemistry efficiency of photosystem II (Fv/Fm), photochemical quenching coefficient, efficient quantum yield, and photosynthetic electron transport rate; and significantly induced the expressions of cold response genes (viz., *CSD1C*, *Why 1*, *HOS1*, and *CBF7-1*)	[152]
**Heavy metal stress**			
*Exophiala pisciphila*	*Z. mays*	More tolerant to cadmium stress; colonized plant root; significantly enhanced plant growth, antioxidants, and antioxidant enzyme activities; altered metal chemical form into an inactive form; repartitioned subcellular cadmium into the cell wall; and bioaugmented cadmium tolerance	[153]
*Gaeumannomyces cylindrosporus*	*Z. mays*	More tolerant to lead stress; colonized plant roots; enhanced plant biomass, height, and basal diameter; improved photosynthesis efficiency; and modified translocation and accumulation of lead in plants	[154]
*Phialophora mustea*	*L. esculentum*	More tolerant to cadmium and zinc stress, colonized plant roots, improved plant growth, enhanced cadmium and zinc stress tolerance, decreased metal uptake accumulation, increased activity of antioxidant enzymes (viz., SOD and POD), relieved membrane lipid peroxidation damage, and reduced leaf malondialdehyde concentration	[155]
*P. indica*	*N. tabacum*	Improved plant cadmium stress tolerance, increased cadmium accumulation in roots, decreased cadmium accumulation in leaves, increased POD activity and glutathione concentration, and significantly upregulated expression of photosynthesis-related proteins, GS and POD	[156]
*Paecilomyces lilacinus*	*S. lycopersicum*	Improved plant cobalt and lead stress tolerance; increased plant growth, weight, sugar, flavonoids, phenols, indole acetic acid, proline, protein, and relative water content in plants; and alleviated damages caused by cobalt and lead stress	[157]
*Rhizoscyphus* sp., *Rhizodermea veluwensis*, and *Phialocephala fortinii*	*Clethra barbinervis*	More tolerant to heavy metal stress; increased seedling growth and K uptake in shoots; and decreased concentration of heavy metals (such as zinc, nickel, lead, copper, and cadmium) in roots	[158]
*Purpureocillium* sp.	*Kandelia candel*	More tolerant to copper stress; protected the plant growth; increased chlorophyll, water saturation deficit, and relative water content in leaves; reduced plant uptake of copper; increased concentration of copper complexes in soil; and reduced copper ion	[159]
*P. roqueforti*	*T. aestivum*	More tolerant to heavy metal stress, secreted indole acetic acid, restricted heavy metal transfer from soil to plants, caused higher plant growth and nutrient uptake, and caused lower level of heavy metals (such as cadmium, copper, lead, nickel, and zinc) in plants	[160]
*Trametes hirsuta*	*T. aestivum*	More tolerant to high lead concentration and increased plant cumulative growth, total chlorophyll content, and lead accumulation in plants	[161]
*E. pisciphila*	*Z. mays*	Improved plant cadmium stress tolerance, colonized plant roots, increased plant biomass and height, induced higher cadmium holding capacity in the root cell wall, and modulated root cell wall with polysaccharide components	[162]

Note: PAL—phenylalanine ammonia lyase; PPO—polyphenol oxidase; POD—peroxidase; GR—glutathione reductase; APX—ascorbate peroxidase; CAT—catalase; GS—glutathione synthase; SOD—superoxide dismutase; AAO—ascorbic acid oxidase; GPX—guaiacol peroxidase; CO_2_—carbon dioxide; *CSD1C*—copper/zinc superoxide dismutase 1C; *Why 1*—transcription factor WHIRLY 1; *HOS1*—high expression of osmotically responsive gene 1; *CBF7-1*—C-repeat-binding factor 7-1.

## Data Availability

Data are contained within the article.

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
