# Peer review of "Fungal Endophytes as Mitigators against Biotic and Abiotic Stresses in Crop Plants"

_jof, 2024, doi:10.3390/jof10020116_

Round 1
Reviewer 1 Report
Comments and Suggestions for Authors
The review by Gowtham et al. is somewhat informative in improving the understanding and application of fungal endophytes in sustainable agricultural systems. However, the following points need to be addressed:
1. In addition to the beneficial effects, the adverse effects of the use of fungal endophytes should also be addressed in detail, except for one sentence in the last part of the paper.
2. The English writing of the text needs to be improved to a more concise style. For example, in lines 41-45, there is a very complicated sentence that covers many points without a main theme. My suggestion is to use more simple sentences and make the logical relationship between the preceding and following sentences more reasonable. This kind of long and complicated sentence appeared throughout the text.
3. Lines 45-47, the meaning of this sentence has already been covered in the text.
4. The authors may have overestimated the importance of fungal endophytes (lines 79-81; lines 90-92); in fact, the beneficial effects of bacterial endophytes on host plants have been of equal or greater concern in both theoretical and practical research. Please be objective about the importance of endophytic fungi.
5. Lines 92-95: Researchers have tended to accept that the definition of endophytes should only refer to their habitat and not include their functions such as asymptom or pathogenicity (see reference: Hardoim PR, van Overbeek LS, Berg G, Pirttil? AM, Compant S, Campisano A, et al. The hidden world inside plants: Ecological and evolutionary considerations for defining the function of microbial endophytes. Microbiology & Molecular Biology Reviews. 2015;79(3):293-320. doi: 10.1128/MMBR.00050-14). Therefore, some expressions in the text should be modified accordingly.
6. Lines 136-137: Unreasonable logical relation to the previous text.
7. Line 140, Line 317 (Figure 1 and Figure 3): It is not appropriate to compare plants with and without endophytic fungi. All plants are exclusively hosts of endophytic fungi and not all colonised fungal endophytes confer beneficial effects on the host, you can refer to plants colonised with or without beneficial fungal endophytes. Figure 3 is less necessary as it gives less information. If the authors wish to summarise the mechanism of fungal endophytes in alleviating abiotic stress, a table will suffice.
8. Many endophytic fungi are latent pathogens, and as the authors have mentioned in the challenges and future aspects section, it is necessary to discuss how these fungal regents can be used appropriately.
Author Response
We profusely thank the reviewr for his constructive comments and hope so the response will be found agrreable.
Comment No. 1: In addition to the beneficial effects, the adverse effects of the use of fungal endophytes should also be addressed in detail, except for one sentence in the last part of the paper.
Response: As per the Reviewer’s suggestion, the adverse effects upon the usage of fungal endophytes on host plants have been addressed in the revised manuscript.
Comment No. 2: The English writing of the text needs to be improved to a more concise style. For example, in lines 41-45, there is a very complicated sentence that covers many points without a main theme. My suggestion is to use more simple sentences and make the logical relationship between the preceding and following sentences more reasonable. This kind of long and complicated sentence appeared throughout the text.
Response: As per the Reviewer’s suggestion, the revised MS has been crosschecked with a native English speaker and corrections have been incorporated accordingly.
Comment No. 3: Lines 45-47, the meaning of this sentence has already been covered in the text.
Response: As per the Reviewer’s suggestion, the sentence has been modified with better clarity in the revised MS.
Comment No. 4: The authors may have overestimated the importance of fungal endophytes (lines 79-81; lines 90-92); in fact, the beneficial effects of bacterial endophytes on host plants have been of equal or greater concern in both theoretical and practical research. Please be objective about the importance of endophytic fungi.
Response: As per the Reviewer’s suggestion, the sentences have been modified with better clarity in the revised MS.
Comment No. 5: Lines 92-95: Researchers have tended to accept that the definition of endophytes should only refer to their habitat and not include their functions such as asymptom or pathogenicity (see reference: Hardoim PR, van Overbeek LS, Berg G, Pirttil? AM, Compant S, Campisano A, et al. The hidden world inside plants: Ecological and evolutionary considerations for defining the function of microbial endophytes. Microbiology & Molecular Biology Reviews. 2015;79(3):293-320. doi: 10.1128/MMBR.00050-14). Therefore, some expressions in the text should be modified accordingly.
Response: As per the Reviewer’s suggestion, due care has been taken in order to depict the definition of endophyte and reference has been quoted accordingly in the revised MS.
Comment No. 6: Lines 136-137: Unreasonable logical relation to the previous text.
Response: As per the Reviewer’s suggestion, the sentence has been removed in the revised MS for better clarity.
Comment No. 7: Line 140, Line 317 (Figure 1 and Figure 3): It is not appropriate to compare plants with and without endophytic fungi. All plants are exclusively hosts of endophytic fungi and not all colonised fungal endophytes confer beneficial effects on the host, you can refer to plants colonised with or without beneficial fungal endophytes. Figure 3 is less necessary as it gives less information. If the authors wish to summarise the mechanism of fungal endophytes in alleviating abiotic stress, a table will suffice.
Response: As per the Reviewer’s suggestion, Figure 1 has been modified accordingly and Figure 3 is removed in the revised MS.
Comment No. 8: Many endophytic fungi are latent pathogens, and as the authors have mentioned in the challenges and future aspects section, it is necessary to discuss how these fungal regents can be used appropriately.
Response: As per the Reviewer’s suggestion, endophytic fungi as latent pathogens have been discussed appropriately in the revised MS in the section “Challenges and future aspects”.
Reviewer 2 Report
Comments and Suggestions for Authors
This publication presents a review on the different aspects of biotic and abiotic stress mitigation in plants by the application of endophytes fungi. The adopted approach in this review was very interesting since endophytes fungi figure among the most significant components of the plant microbiome and could be a promising asset to boost plant tolerance to different stresses.
The manuscript was well introduced, and the authors shed lights on the mechanisms orchestrating the fungal endophyte’s role in the alleviation of environmental stress on crops. However, the manuscript needs major revisions to be suitable for publication in Journal of Fungi.
General comments
- The English of this manuscript needs moderate improvement (grammatical and punctuation checks).
- Please provide some examples of (up/downregulated) genes involved in the improvement of plant tolerance to environmental stresses when inoculated with fungal endophytes. Please include examples in each subsection dealing with this aspect.
- Please provide a subsection on the interactions between fungal endophytes and other members of the plant microbiome since their effects on crop tolerance to environmental stress is altered by other members of this microbiome such as PGPR, AMF, protists, archea....
- Please provide, at the end of each section/subsection, the new avenues to develop regarding the discussed aspect.
- Please provide a subsection on the role of fungal endophytes in mitigating chilling stress.
Other comments
- L62: please change “distributed variably” to “variably distributed”. Please check throughout the manuscript.
- L197: please provide the significance of the abbreviations at the first appearance in the text. Please check throughout the manuscript.
- L203: please do not italicize “(FlSp1)”.
- Tables 1 and 4: please provide the significance of all the abbreviations included in this table as a footnote.
Comments on the Quality of English Language
The English of this manuscript needs moderate revision.
Author Response
We profusely thank the reviewr for his constructive comments and hope so the response will be found agrreable.
General Comments
Comment No. 1: The English of this manuscript needs moderate improvement (grammatical and punctuation checks).
Response: As per the Reviewer’s suggestion, the revised MS has been crosschecked with a native English speaker and corrections have been incorporated accordingly.
Comment No. 2: Please provide some examples of (up/downregulated) genes involved in the improvement of plant tolerance to environmental stresses when inoculated with fungal endophytes. Please include examples in each subsection dealing with this aspect.
Response: As per the Reviewer’s suggestion, information related up and down regulation of genes upon inoculation with endophytic fungi under stress is included in the revised MS (Table 4).
Comment No. 3: Please provide a subsection on the interactions between fungal endophytes and other members of the plant microbiome since their effects on crop tolerance to environmental stress is altered by other members of this microbiome such as PGPR, AMF, protists, archea....
Response: As per the Reviewer’s suggestion, a subsection on the interaction between fungal endophytes and other plant microbiome has been incorporated in revised manuscript.
Comment No. 4: Please provide, at the end of each section/ subsection, the new avenues to develop regarding the discussed aspect.
Response: As per the Reviewer’s suggestion, the new avenues related to each section/ subsection is included in the revised MS.
Comment No. 5: Please provide a subsection on the role of fungal endophytes in mitigating chilling stress.
Response: As per the Reviewer’s suggestion, a subsection on the role of fungal endophytes in mitigating chilling/ cold stress has been incorporated in revised manuscript.
Other comments:
Comment No. 6: L62: please change “distributed variably” to “variably distributed”. Please check throughout the manuscript.
Response: As per the Reviewer’s suggestion, the correction has been made accordingly in revised manuscript.
Comment No. 7: L197: please provide the significance of the abbreviations at the first appearance in the text. Please check throughout the manuscript.
Response: As per the Reviewer’s suggestion, the full form of all the abbreviation has been provided in full form and later abbreviated accordingly in the revised MS.
Comment No. 8: L203: please do not italicize “(FlSp1)”.
Response: As per the Reviewer’s suggestion, the correction has been made accordingly in revised manuscript.
Comment No. 9: Tables 1 and 4: please provide the significance of all the abbreviations included in this table as a footnote.
Response: As per the Reviewer’s suggestion, all the abbreviations have been included accordingly in the Tables 1 and 4 as a footnote in revised manuscript.
Round 2
Reviewer 2 Report
Comments and Suggestions for Authors
The authors satisfied all the raised comment and I endorce the publication of the current version of the manuscript.